# Novel protein markers of androgen activity in humans: proteomic study of plasma from young chemically castrated men

Aleksander Giwercman[1]*, K Barbara Sahlin[2,3], Indira Pla Parada[2,3], Krzysztof Pawlowski[2,4,5], Carl Fehninger[2,3], Yvonne Lundberg Giwercman[6], Irene Leijonhufvud[1], Roger Appelqvist[2,3], György Marko-Varga[3,7], Aniel Sanchez[2,3†], Johan Malm[2,3†]

[1]Molecular Reproductive Medicine, Department of Translational Medicine, Lund University, Malmo, Sweden; [2]Section for Clinical Chemistry, Department of Translational Medicine, Lund University, Skåne University Hospital Malmö, Lund, Sweden; [3]Clinical Protein Science & Imaging, Biomedical Centre, Department of Biomedical Engineering, Lund University, Lund, Sweden; [4]Department of Experimental Design and Bioinformatics, Faculty of Agriculture and Biology, Warsaw University of Life Sciences SGGW, Warszawa, Poland; [5]Department of Molecular Biology, University of Texas Southwestern Medical Center, Dallas, United States; [6]Molecular Genetic Reproductive Medicine, Department of Translational Medicine, Lund University, Lund, Sweden; [7]First Department of Surgery, Tokyo Medical University, Nishishinjiku Shinjiku-ku, Japan

*For correspondence:
aleksander.giwercman@med.
lu.se

†These authors contributed equally to this work

## Abstract

**Background:** Reliable biomarkers of androgen activity in humans are lacking. The aim of this study was, therefore, to identify new protein markers of biological androgen activity and test their predictive value in relation to low vs normal testosterone values and some androgen deficiency linked pathologies.

**Methods:** Blood samples from 30 healthy GnRH antagonist treated males were collected at three time points: (1) before GnRH antagonist administration; (2) 3 weeks later, just before testosterone undecanoate injection, and (3) after additional 2 weeks. Subsequently, they were analyzed by mass spectrometry to identify potential protein biomarkers of testosterone activity. Levels of proteins most significantly associated with testosterone fluctuations were further tested in a cohort of 75 hypo- and eugonadal males suffering from infertility. Associations between levels of those markers and cardiometabolic parameters, bone mineral density as well as androgen receptor (AR) CAG repeat lengths, were explored.

**Results:** Using receiver operating characteristic analysis, 4-hydroxyphenylpyruvate dioxygenase (4HPPD), insulin-like growth factor-binding protein 6 (IGFBP6), and fructose-bisphosphate aldolase (ALDOB), as well as a Multi Marker Algorithm, based on levels of 4HPPD and IGFBP6, were shown to be best predictors of low (<8 nmol/l) vs normal (>12 nmol/l) testosterone. They were also more strongly associated with metabolic syndrome and diabetes than testosterone levels. Levels of ALDOB and 4HPPD also showed association with AR CAG repeat lengths.

**Conclusions:** We identified potential new protein biomarkers of testosterone action. Further investigations to elucidate their clinical potential are warranted.

**Funding:** The work was supported by ReproUnion2.0 (grant no. 20201846), which is funded by the Interreg V EU program.

## Editor's evaluation

This work by Giwercman, et al., interrogates the identity of potential protein biomarkers of androgen activity in humans by carrying out a proteomic analysis in blood from 30 healthy males treated at baseline, after medical castration and at a third time point after testosterone replacement. Proteins most significantly associated with testosterone changes were tested further in a separate cohort. The major findings include the observation that 4 specific proteins are potential protein biomarkers that follow testosterone levels and presumably androgen receptor activity, thus providing new insights into androgen physiology and pathophysiology.

## Introduction

The male sex hormone, testosterone (T), plays an important physiological role in regulating function of both reproductive and nonreproductive organs in males as well as in females. In males, the diagnosis of T deficiency (i.e., male hypogonadism) is based on the presence of low serum T levels combined with clinical symptoms, which, however, are not pathognomonic for this condition (*Traish et al., 2011*).

The most common way of assessing T activity is by measuring the total concentration of this hormone in a fasting morning blood sample. However, total T does not accurately reflect biological androgenic activity (BAA), which might be considered a more useful biological and clinical marker.

The association between T levels and BAA is affected by several biological mechanisms such as the concentration of binding proteins, body mass index, certain diseases (e.g., diabetes), androgen receptor (AR) sensitivity (*Zitzmann, 2008*), and different cofactors (*Furuya et al., 2013*). So far, no reliable algorithms for translating T levels into BAA are available, but could be useful for example in the diagnosis of male hypogonadism.

A correct hypogonadism diagnosis is important for proper identification of men for whom androgen replacement therapy is warranted. However, the treatment of men with hypogonadism represents a clinical challenge, because the symptoms associated with the condition are highly nonspecific (*Miner et al., 2014*). Furthermore, there are limitations in using the level of T in defining testosterone deficiency. Generally, in many clinical guidelines, total T concentration below 8 nmol/l indicates an insufficient hormone concentration, whereas levels above 12 nmol/l are considered normal (*Arver and Lehtihet, 2009*). Apart from the fact that men with testosterone levels between 8 and 12 nmol/l cannot be assigned to any of these distinct categories, those presenting with a lower or higher hormone concentration may also be misclassified due to an abnormal concentration of sex hormone-binding protein (SHBG). Low SHBG levels, as often seen in obese men, may imply low total T despite unaffected BAA. On the other hand, some degree of reduced androgen sensitivity may be associated with decreased BAA despite normal or high testosterone levels (*Diaz-Arjonilla et al., 2009*).

Hypogonadism has been identified as a predictor of several noncommunicable chronic diseases as well as premature mortality (*Muraleedharan and Jones, 2014*). Understanding the biology of androgen action may therefore contribute to clarifying the pathogenetic mechanisms linking androgen deficiency to comorbid conditions. Thus, the approach based on measuring the protein levels downstream of androgen action is a feasible and logical concept for identifying clinical and biological useful markers of BAA.

Proteomics is a technique aimed to study biological systems based on qualitative and quantitative measuring of proteins and, thereby, integrate the cellular output related to transcription as well as translation. Mapping the quantitative protein response downstream of androgen action may provide new clinically valuable markers of BAA. In order to identify such markers, we compared the protein profile of healthy individuals before and after T deprivation. Subsequently, we assessed the predictive value of the newly identified protein markers in relation to hypogonadism and risk of pathologies related to T deficiency.

**eLife digest** Although it is best known for its role in developing male sex organs and maintaining sexual function, the hormone testosterone is important for many parts of the human body. A deficiency can cause an increased risk of serious conditions such as diabetes, cancer and osteoporosis. Testosterone deficiency can develop due to disease or age-related changes, and men affected by this can be given supplements of this hormone to restore normal levels.

The most common way to test for testosterone deficiency is by measuring the concentration of the hormone in the blood. However, this does not accurately reflect the activity of the hormone in the body. This may lead to men who need more testosterone not receiving enough, and to others being unnecessarily treated. Several factors may lead to discrepancy between testosterone concentration in blood and its physiological activity. One of the most common is obesity. Additionally, certain genetic factors, which cannot be controlled for yet, regulate sensitivity to this hormone: some people do well at low levels, while others need high concentrations to be healthy. Therefore, to improve the diagnosis of testosterone deficiency it is necessary to identify biological markers whose levels act as a proxy for testosterone activity.

Giwercman, Sahlin et al. studied the levels of a large number of proteins in the blood of 30 young men before and after blocking testosterone production. The analysis found three proteins whose concentrations changed significantly after testosterone deprivation. Giwercman, Sahlin et al. then validated these markers for testosterone deficiency by checking the levels of the three proteins in a separate group of 75 men with fertility problems. The results also showed that the three protein markers were better at predicting diabetes and metabolic syndrome than testosterone levels alone.

These newly discovered markers could be used to create a test for measuring testosterone activity. This could help to identify deficiencies and finetune the amount of supplementary hormone given to men as treatment. However, further research is needed to understand the clinical value of such a test in men, as well as women and children.

## Materials and methods

### Study outline

The study was set up to (1) identify new protein markers of BAA in healthy subjects; (2) test the markers' predictive values in relation to biochemically diagnosed hypogonadism, metabolic syndrome (MetS), cardiovascular risk lipid profile (CVRLP), diabetes mellitus II (DM), and low bone density (LBD) in infertile men; (3) analyze androgen dependence of the identified proteinsby assessing how their levels associate with AR gene CAG repeat length.

### Subjects

All subjects were enrolled with informed written consent. The two studies from which they were recruited were approved by the Swedish Ethical Review Authority (approval number: DNR 2014/311, date of approval May 8, 2014; DNR 2011/1, date of approval January 11, 2011).

The first part of the study includes plasma samples obtained from 30 healthy men (biological replicates) aged 19–32 years, BMI 19.1–26.9 kg/m². They underwent chemical castration by subcutaneous administration of 240 mg GnRH antagonist (Degeralix, Ferring Pharmaceuticals, Saint-Prex, Switzerland) followed by remediation of testosterone levels by intramuscular injection of 1000 mg testosterone undecanoate (Nebido, Bayer AG, Leverkusen, Germany) after the duration of 3 weeks (*Sahlin et al., 2020*; *Pla et al., 2020*). Blood samples were collected at baseline (A), 3 weeks later (B), and, at the end of the study, after two additional weeks (C) (*Figure 1*).

To test the clinical predictive value of the proteins identified in the castrated men, we used a cohort of 75 serum samples from 75 men (biological replication, subject age 32–43 years) previously recruited for a study on hypogonadism among men from infertile couples (*Bobjer et al., 2016*). Eighty-five patients were randomly selected from 213 infertile men and 223 age-matched controls. The selected patients for the present study had the span of subnormal to upper normal range of T. Out of the 85 patients, 10 patients were excluded; 7 due to Klinefelter syndrome, 1 due to missing value of T, and the last 2 were statistical outliers, which were removed after considering the possible causes. One

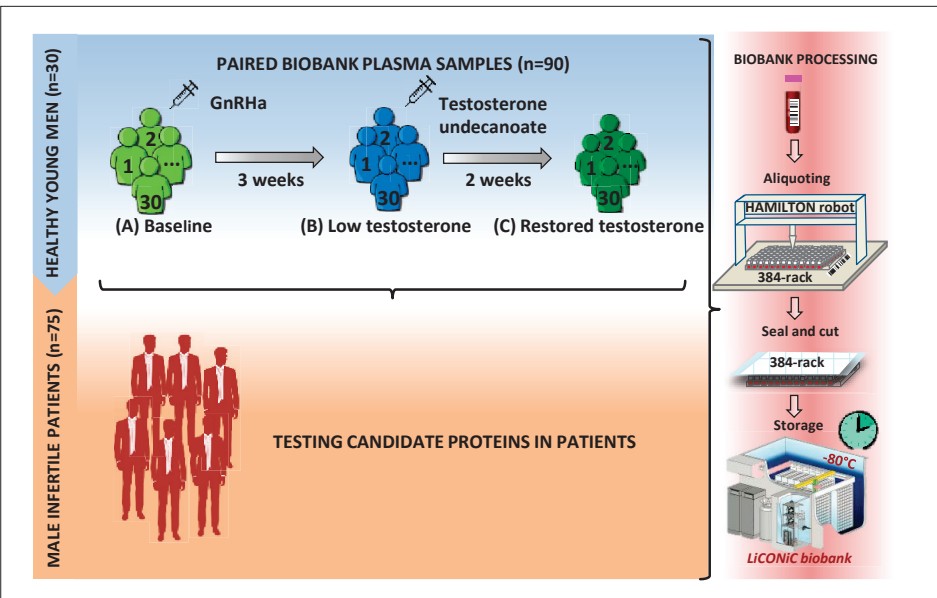

**Figure 1.** Study design. First, a model of 30 young healthy males was evaluated by proteomics at three time points (**A–C**), where testosterone changes were induced: (**A**) baseline; (**B**) week 3; (**C**) week 5. Identified proteins proposed as candidate biomarkers were then evaluated in a cohort of infertile males. In both steps of the study, the quality of the blood samples was ensured by following an automated workflow for sample aliquoting and storage (−80°C).

patient had a high level of T (41.6 nmol/l) without androgen replacement therapy and the other because he was the only one diagnosed with obstructive azoospermia. Background characteristics of these patients can be found in *Table 1a, b*.

The following comorbidities in the cohort of infertile patients were defined MetS, IR, CVRLP, DM, and LBD. MetS was determined according to the criteria defined at the National Cholesterol Education Program Adult Treatment Panel III 2002 (in Table S1 available here). Homeostatic Model Assessment of Insulin Resistance (HOMA-IR) was calculated as (glucose × insulin)/22.5 and IR was defined as HOMA-IR >2.5 (*Wickramasinghe et al., 2017*). CVRLP was defined as the ratio apolipoprotein B/apolipoprotein A1 >0.9 (*Walldius and Jungner I, 2004*). DM was set at fasting blood glucose >7 mmol/l (*American Diabetes Association, 2010*). LBD was determined based on the DEXA lumbar z-score with the cutoff at <−1 (*Isaksson et al., 2017*). The methods for laboratory tests (*Bobjer et al., 2016*), CAG repeat length (*Lundin et al., 2003*), and proteomics (*Smith et al., 1985*; *MacLean et al., 2010*) are described in the supplementary Appendix 1.

## Statistical analysis

We briefly describe the statistical analyses performed. A full description of the statistical analysis is available in Appendix 1 – *Supplementary Statistical analyses*. Proteomics data preprocessing was done using Perseus v1.6.7.0 (*Tyanova et al., 2016*) software and unless other software is specified, the statistical analyses were performed using R software (*RStudio Team, 2016*; *R Development Core Team, 2016*).

### Healthy human model

Protein intensities were Log2 transformed and standardized by Subtract Median normalization. Differentially expressed proteins were determined by one-way repeated measures ANOVA followed by a pairwise t-test (two tails and paired). Adjusted p values <0.05 were considered sf were considered candidate biomarker. These candidates were include significant. The power of the candidate biomarkers to discriminate between normal and low T was evaluated by doing receiver operating characteristic (ROC) analysis. Significant proteins (between A and B with significant recovery in B and C) with (1) area under the curve (AUC) >0.80 or (2) AUC between 0.75 and 0.80 (*Marshall et al., 2010*; *Bowers and Zhou, 2019*; *Simundić, 2009*) and highly enriched in liver tissues Human Proteome Map (*Kim et al., 2014*) and (*Kampf et al., 2014*; *Kholodenko and Yarygin, 2017*; *Schmucker and*

**Table 1.** Background characteristics of the infertile patients.

a. **Background characteristics of infertile patients.**

|  | Mean (SD) | *N* |
|---|---|---|
| Age at inclusion (years) | 37.8 (5.5) | 75 |
| BMI | 27.2 (4.3) | 72 |
| Total testosterone (nmol/l) | 12.8 (6.8) | 75 |
| FSH (IU/l) | 15.8 (14.3) | 75 |
| LH (IU/l) | 7.5 (5.7) | 75 |
| SHBG (nmol/l) | 24.0 (4.5–84.5)* | 75 |
| Estradiol (pmol/L) | 96 (36–321)* | 75 |
| Calculated free testosterone (pmol/l) | 260 (50–1360)* | 75 |
| ApoB/ApoA1 | 0.7 (0.2) | 68 |
| HOMA-IR | 1.6 (0.4–13.9)* | 75 |
| DEXA score (lumbar *z*-score) | −0.5 (1.3) | 74 |
| CAG (repeated length) | 22 (14–31)* | 74 |

b. Characteristics of the cohort of infertile patients

|  | *n* (%) |
|---|---|
| Smoker | 9 (12.0) |
| Current diseases | 36 (48.0) |
| Insulin medication | 1 (1.3) |
| Current ART | 8 (10.7) |
| *CVRLP* | 20 (27.0) |
| Insulin resistance | 20 (27.0) |
| Diabetes mellitus 2 | 4 (5.3) |
| Metabolic syndrome (MetS) | 14 (20.9) |
| Low bone density | 23 (30.6) |

Variables are expressed as frequency (*n*) and percentage of occurrence (%) – according to the total number of patients (*N* = 75).
ART: androgen replacement therapy;
CVRLP: cardiovascular risk lipid profile.
*Nine with allergy; eight with local pains; five psychiatric disorders, four hormonal deficiencies; two with diabetes, and eight with mixed conditions.

*Characteristics values are expressed as mean (SD), except for those that did not follow a normal distribution (non-Gaussian) and which are shown as median (min–max).

*Sanchez, 2011*) were considered candidate biomarker. These candidates were included as predictors in a stepwise regression (method: backward) to select the best combination of markers that predict the odds of being low T. Bootstrap resampling with replacement method was applied to assess consistency. A new variable called Multi Marker Algorithm (MMA) was derived from the predicted log-odds (of being low T) obtained from a binomial logistic regression analysis (see Appendix 1 – *Supplementary Statistical analyses*) and it was evaluated together with marker candidates proteins.

## Infertile cohort of patients

The normal distribution of the variables that describe background characteristics of the infertile cohort of patients (*Table 1*) was evaluated by Kolmogorov–Smirnov test. The intensities of the candidate biomarkers were Log2 transformed to achieve normal distributions. In this cohort, MMA variable was created to predict the odds of suffering low T or other medical conditions associated with low T levels.

Changes between two groups were evaluated by two-tailed Student's *t*-test (p values <0.05 were considered significant). Overall changes between more than two groups were evaluated by one-way ANOVA followed by a pairwise FDR correction (*Benjamini et al., 2006*) and adjusted p values <0.05 were considered significant. In order to know if the changes in the candidate markers occur with the change in T as observed in the healthy human model, three groups of patients were created based on total T concentration (*Arver and Lehtihet, 2009*) (group 1: low T [LT] ≤8 nmol/l [*n* = 22]; group 2: borderline low T [BL_T] between 8 and 12 nmol/l [*n* = 17]; group 3: normal T [NT] >12 nmol/l [*n* = 36]). Calculated free testosterone (cFT) was determined according to the method described by *Vermeulen et al., 1999*. The cutoff level of 220 pmol/l was used to categorize the subjects as having low cFT (L_cFT; *n* = 21) or normal cFT (N_cFT; *n* = 54) (*Antonio et al., 2016*).

The power of the candidate biomarkers to discriminate between LT and NT (including or not the BL_T) (*Chan et al., 2014*; *Lunenfeld et al., 2012*; *Zitzmann et al., 2006*), or to distinguish patients with medical conditions associated with low T levels (MetS, IR, CVRLP, DM, or LBD) was evaluated by an ROC analysis. The same was done to discriminate between L_cFT and N_cFT. The DeLong's test (paired) was used to compare the AUCs.

In order to strengthen the evidence of androgenic dependence of the candidate biomarkers, we looked for associations between their expression and the AR CAG repeat length, which was previously reported to have an impact on the activity of the receptor (*Casella et al., 2001*; *Stanworth et al., 2008*; *Kim et al., 2018*; *Ferlin et al., 2004*). Three categories were defined: reference group 1: patients with CAG repeat length 21 and 22 (*n* = 18); group 2: patients with CAG repeat length <21 (*n* = 26), and group 3: patients with CAG repeat length >22. This categorization was undertaken in order to have three groups of sufficient size and the category including the mean CAG length value of 22 was chosen as reference since this CAG number was previously seen, in vitro and in vivo to be associated with highest receptor activity (*Nenonen et al., 2010*; *Nenonen et al., 2011*).

# Results

## Proteins differentially expressed in chemically castrated men

In total, in the healthy men, the expression level of 31 out of 676 proteins was statistically significantly associated with T concentration (in Table S2 available here). The levels of 23 proteins changed in the same direction as T, whereas, the remaining eight markers changed in an opposite way. LH and FSH changed significantly in A and B but not in B and C. The protein changes visualized as boxplots can be found in Figure 2—figure supplement 1 available at https://doi.org/10.6084/m9.figshare.14876562.

## Proteins capable to distinguish between low and normal testosterone

Based on p values for AUC in the ROC analysis, among healthy young men, 90% of the 31 proteins distinguished the low T time point (B) from the normal ones (A and C) with statistical significance(in Table S3 available here, *Figure 2a*). ROC–AUC values greater than 0.80 were obtained for the proteins 4-hydroxyphenylpyruvate dioxygenase (4HPPD) and insulin-like growth factor-binding protein 6 (IGFBP6). Additionally, fructose-bisphosphate aldolase (ALDOB) was the only protein enriched in liver tissue with ROC–AUC between 0.75 and 0.80.

The stepwise regression method selected 4HPPD and IGFBP6 as the best markers to be combined to predict the odds of being low T, and thus, they were the basis for the new variable MMA (see Material and methods). MMA together with 4HPPD, ALDOB, and IGFBP6 was selected as potential candidate markers for the diagnosis of BAA (*Figure 2b, c*). The expression of the 4HPPD and ALDOB proteins was significantly increasedat low T (p < 0.001; p < 0.001) and remediated in response to the T treatment, whereas IGFBP6 expression was significantly decreased (p < 0.001) by castration.

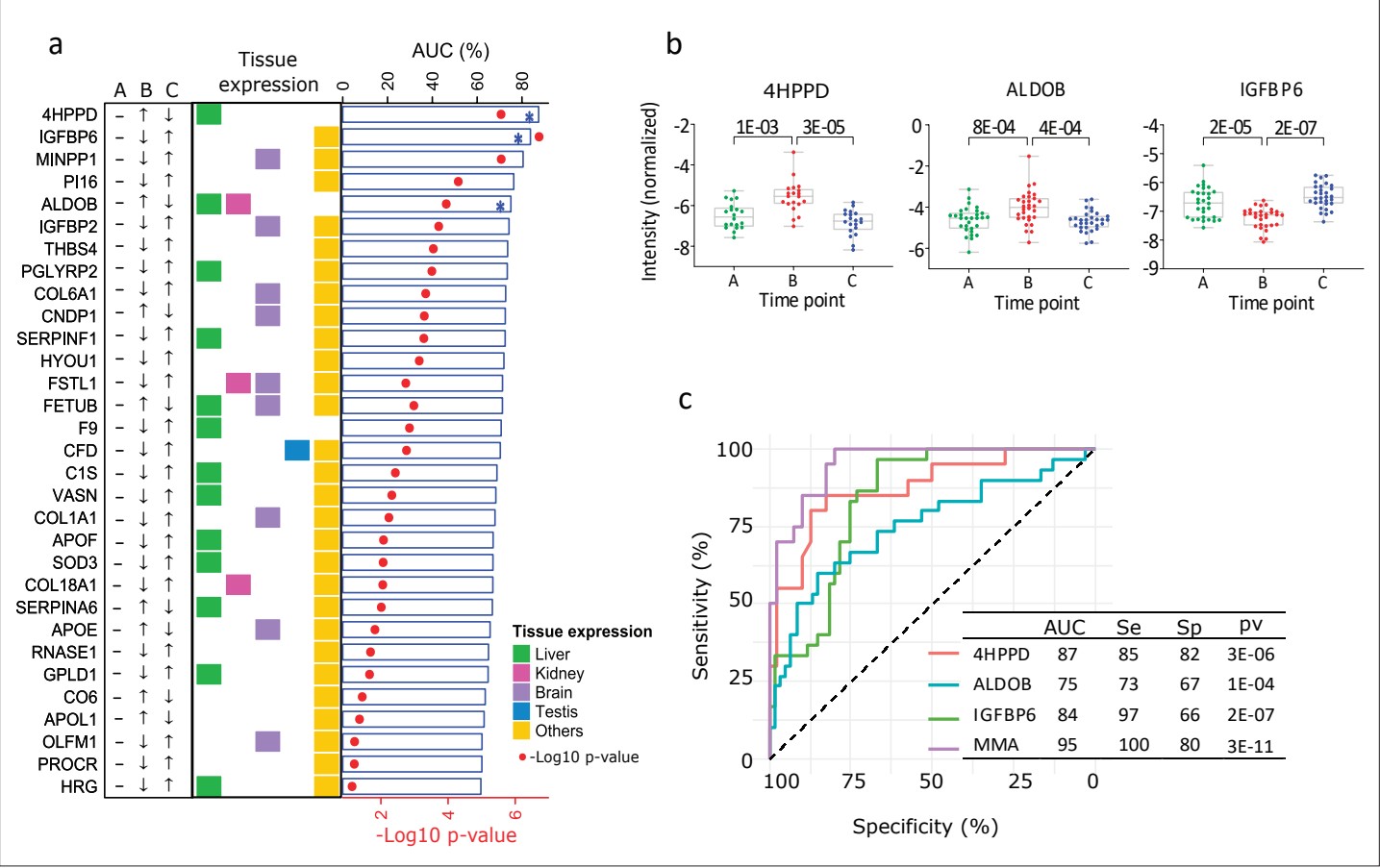

**Figure 2.** Proteins influenced by testosterone in the model of young healthy males. (**a**) Top 25 significant proteins selected in the healthy human model (receiver operating characteristic [ROC] p < 0.01, Table S3). The arrows indicate the direction of change in protein expression in the different conditions. The tissue with highest expression of each protein is indicated in colors. Also, results from the ROC analysis are shown as bar chart (area under the curve, AUC) and heat-map (p values). (**b**) Boxplot (mean (min; max)) of the top three significant proteins proposed as biomarker candidates, able to discriminate between low and normal testosterone (in Table S4 available here). The adjusted p values are specified on top of the comparative horizontal lines. (**c**) ROC of the analytes proposed as biomarker candidates, including Multi Marker Algorithm (MMA).

## Testing of the candidate biomarkers in infertile men

The three proteins and MMA showed statistically significant differences (4HPPD: p < 0.001, ALDOB: p = 0.003, IGFBP6: p = 0.016, MMA: p < 0.001, *Figure 3a*) between the three groups defined according to the total T levels (in Table S2 available here). 4HPPD, ALDOB, and MMA showed a negative association with T changes, while IGFBP6 displayed a positive association. The three proteins and MMA significantly distinguished the patients with LT from BL_T/NT (4HPPD: AUC = 0.75, p = 0.001; ALDOB: AUC = 0.70, p = 0.008; IGFBP6: AUC = 0.69, p = 0.01; MMA: AUC = 0.79, p < 0.001) (*Figure 3b*). Additionally, the power to discern low T values improved for all the biomarkers tested when the patients with BL_T were excluded (*Table 2*). Similar results were obtained for discrimination between low and normal FT (*Figure 3d, e*; *Table 2*).

## Ability to distinguish men with abnormal metabolic comorbidities or reduced bone mineral density

The AUCs for 4HPPD, ALDOB, and MMA in relation to risk of DM and MetS were statistically significant whereas for T the p value for AUC was 0.72. The AUCs for 4HPPD, ALDOB, and MMA were also statistically significantly larger than this for T (DM: 4HPPD [p = 0.005], ALDOB [p = 0.009], MMA [p = 0.002]; MetS: 4HPPD [p = 0.032], ALDOB [p = 0.030], MMA [p = 0.002]; *Table 2*). Additionally, the AUC values in relation toCVRLP and IR were numerically higher for 4HPPD, ALDOB, and MMA than for T, however, the differences between the AUC values were not statistically significant (CVRLP

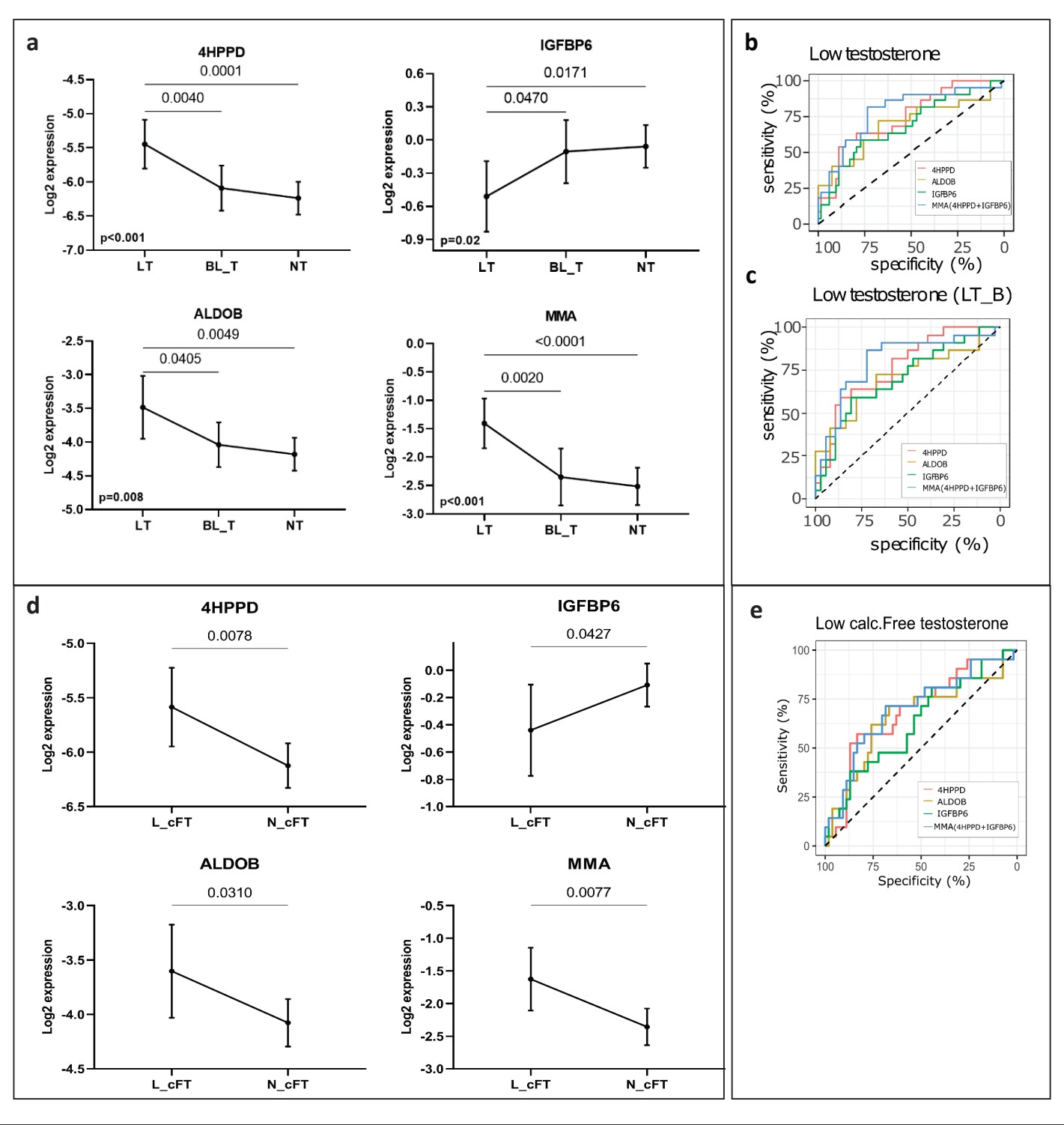

**Figure 3.** New markers to discern states of different testosterone levels in men investigated for infertility (*n* = 75). (**a**) Patients grouped by three levels of total testosterone: low testosterone (LT) ≤8 nmol/l (*n* = 22), borderline testosterone (BL_T) between 8 and 12 nmol/l (*n* = 17) and normal testosterone (HT) >12 nmol/l (*n* = 36). Each group is represented by the mean and its 95% CI. Horizontal lines indicate significant differences between groups and the adjusted p values are specified on top of these lines (in Table S5 available here). (**b**) Receiver operating characteristic (ROC) analysis to discriminate patients with LT in the entire cohort and (**c**) in a cohort that excluded patients with borderline testosterone levels (LT_B). Multi Marker Algorithm (MMA) is based on is the combination of levels of the proteins 4-hydroxyphenylpyruvate dioxygenase (4HPPD) and insulin-like growth factor-binding protein 6 (IGFBP6). (**d**) As (a), but grouped according to the levels of calculated free testosterone (cFT): low (L_cFT) (*n* = 21) − < 220 pmol/l and normal (N_cFT) (*n* = 54) add symbol 220 pmol/l. (**e**) As (b) and (c) but for discrimination of L_cFT and N_cFT.

**Table 2.** Comparison of receiver operating characteristic (ROC)–areas under the curve for testosterone and the candidate biomarkers in relation to the prediction of hypogonadism and its sequelae in patients.

| Analyte | Low T (T ≤ 8 nmol/l) AUC (Sp,Se) | p | Low T* (T ≤ 8 nmol/l) AUC (Sp,Se) | p | Low cFT (cFT <220 pmol/l) AUC (Sp,Se) | p | IR (HOMA-IR >2.5) AUC (Sp,Se) | p | DM - AUC (Sp,Se) | p | LBD (z-score <−1) AUC (Sp,Se) | p | CVRLP (ApoB/ApoA1 ≥0.9) AUC (Sp,Se) | p | MetS AUC (Sp,Se) | p |
|---|---|---|---|---|---|---|---|---|---|---|---|---|---|---|---|---|
| 4HPPD | 0.75 (85,59) | **8.38E−04** | 0.77 (86,59) | **2.66E−04** | 0.69 (83,57) | **5.14E−03** | 0.79 (84,70) | **1.20E−04** | 0.89 (93,75) | **9.00E−03** | 0.64 (76,61) | **2.24E−02** | 0.74 (46,90) | **5.79E−03** | 0.74 (95,50) | **5.31E−03** |
| ALDOB | 0.69 (68,73) | **8.25E−03** | 0.70 (67,73) | **5.39E−03** | 0.66 (67,71) | **1.56E−02** | 0.73 (71,75) | **2.85E−03** | 0.85 (63,100) | **1.80E−02** | 0.57 (98,27) | 1.93E−01 | 0.71 (83,55) | **4.64E−02** | 0.74 (82,64) | **6.02E−03** |
| IGFBP6 | 0.69 (77,59) | **1.05E−02** | 0.70 (81,59) | **4.89E−03** | 0.63 (44,81) | 7.24E−02 | 0.57 (42,80) | 3.50E−01 | 0.59 (39,100) | 5.40E−01 | 0.63 (43,78) | 2.75E−01 | 0.59 (48,80) | 1.87E−01 | 0.65 (45,93) | 8.45E−02 |
| Testosterone | - | - | - | - | - | - | 0.71 (76,70) | **4.96E−03** | 0.55 (32,100) | 7.24E−01 | 0.75 (74,74) | **5.12E−04** | 0.66 (65,85) | **6.55E−03** | 0.56 (72,57) | 5.08E−01 |
| MMA | 0.79 (74,82) | **9.23E−05** | 0.80 (72,86) | **3.86E−05** | 0.70 (69,71) | **3.90E−03** | 0.79 (82,70) | **1.46E−04** | 0.92 (84,100) | **5.00E−03** | 0.78 (82,65) | **1.40E−02** | 0.75 (73,75) | **3.65E−03** | 0.78 (63,86) | **1.57E−03** |

Significant p values are highlighted in bold and underlined. *Excluding patients with testosterone values from the borderline low testosterone (8 < BL_T ≤ 12).

cFT: calculated free testosterone. IR: insulin resistance. DM: diabetes mellitus type 2. LBD: low bone density. CVRLP: cardiovascular risk lipid profile .MetS: metabolic syndrome.

AUC:area under the curve. Spe: specificity in %. Se: sensitivity in %.

marker vs T: 4HPPD [p = 0.97], ALDOB [p = 0.61], MMA [p = 0.87]; IR marker vs T: 4HPPD vs T [p = 0.30], ALDOB vs T [p = 0.88], MMA vs T [p = 0.31]). 4HPPD and MMA statistically significantly distinguished between patients with normal bone density and LBD. The same was true for T, the differences between the AUC for T and those for 4HPPD and MMA not being statistically significant (LBD vs T: 4HPPD [p = 0.28], MMA [p = 0.30]). No statistical significance, in relation to prediction of LBD was found for IGFBP6 (*Figure 4*).

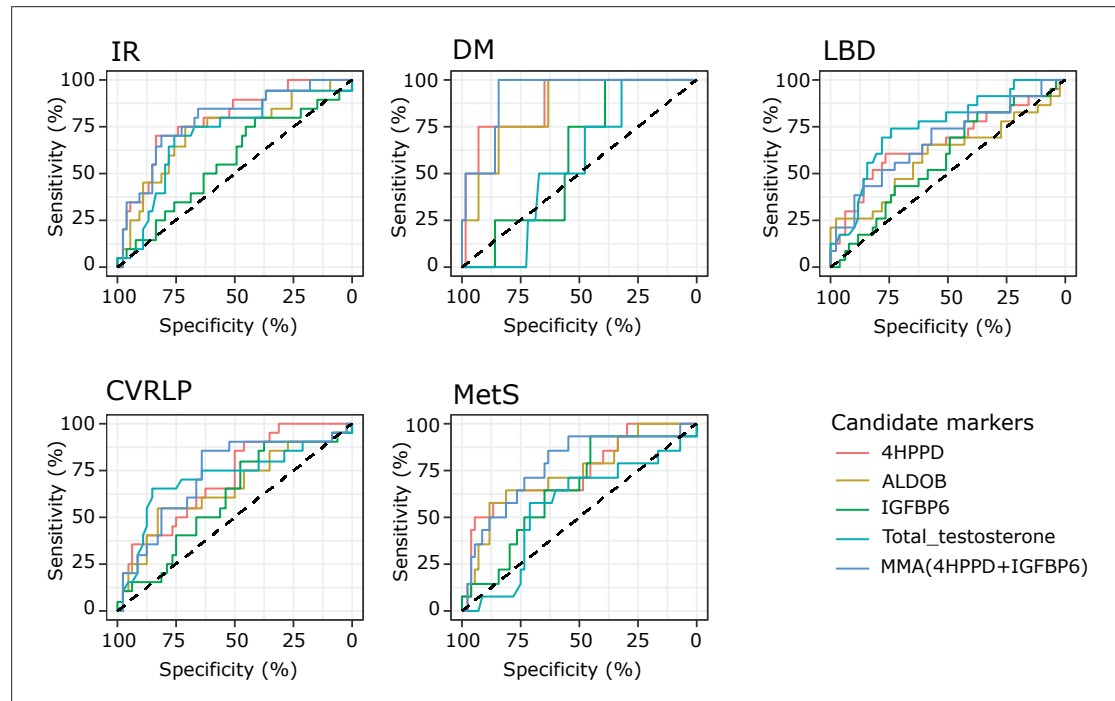

**Figure 4.** Results from receiver operating characteristic (ROC) analysis to determine whether the analytes discriminate between the presence of comorbidities or not. Analytes included in the analysis are 4-hydroxyphenylpyruvate dioxygenase (4HPPD), insulin-like growth factor-binding protein 6 (IGFBP6), fructose-bisphosphate aldolase (ALDOB), and Multi Marker Algorithm (MMA; combination of 4HPPD and IGFBP6). Area under the curve (AUC), p values can be found in *Table 2*. IR: insulin resistance; DM: type 2 diabetes mellitus; LBD: low bone density; CVRLP: cardiovascular risk lipid profile; MetS: metabolic syndrome.

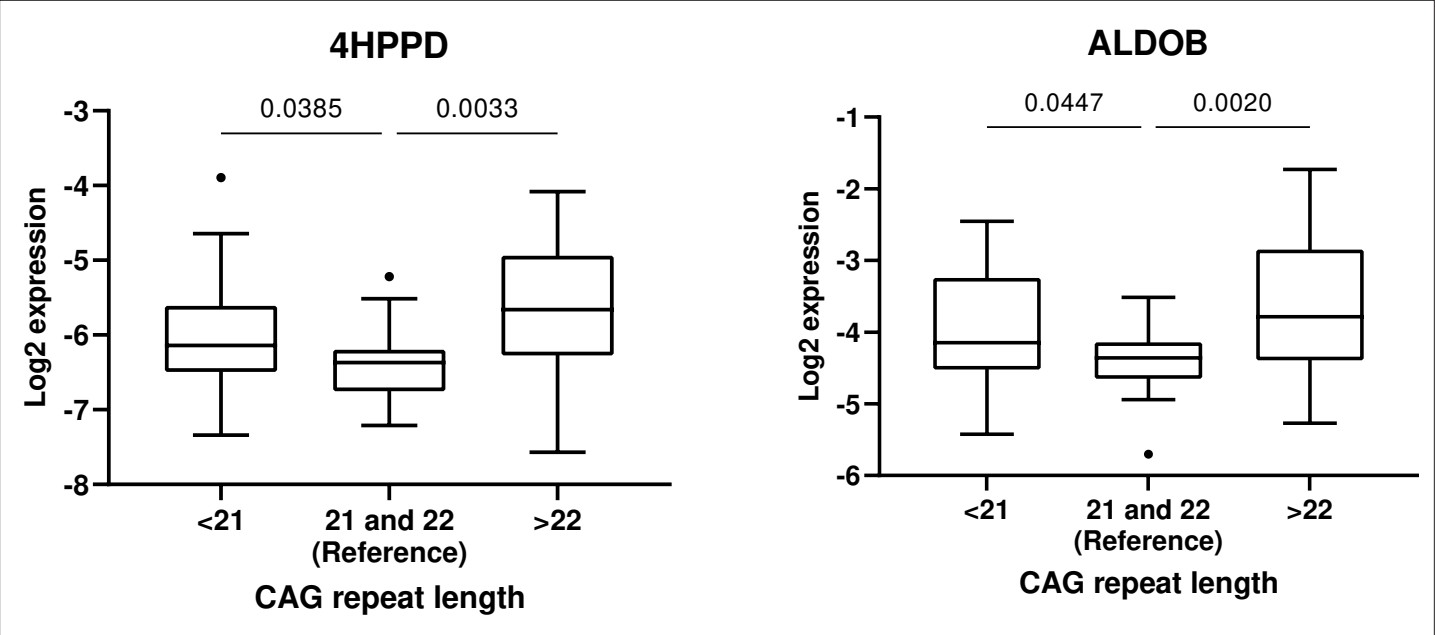

**Figure 5.** Association between androgen receptor CAG <21 (*n* = 26) and CAG <22 (*n* = 30) and 4-hydroxyphenylpyruvate dioxygenase (4HPPD) and fructose-bisphosphate aldolase (ALDOB), respectively, with CAG = 21 and 22 (*n* = 18) set as reference.

### Association of the candidate biomarkers with AR CAG repeat length

Statistically significant inter-CAG-group overall differences were observed for 4HPPD (p = 0.012) and ALDOB (p = 0.008) (*Figure 5*). Additionally, the protein expressions were significantly higher in the groups with <21 and >22 CAG repeat length as compared with the reference (*Figure 5*, *Table 3*). However, we did not observe any statistically significant association between CAG number and expression of IGFBP6.

## Discussion

We have identified three plasma proteins, which are potential markers of BAA. In young healthy men, the three markers ALDOB, 4HPPD, and IGFB6 were strongly associated with T levels. In a slightly older cohort of infertile men, these markers were indicative of T deficiency that is both total and free serum T was low (*Arver and Lehtihet, 2009*). Furthermore, levels of two of the markers, ALDOB and 4HPPD, were more strongly associated with risk of metabolic disturbances than total T. The association seemed to become stronger by creating a combined marker MMA, based on both 4HPPD and IGFBP6 levels. Finally, the androgen dependence of ALDOB and 4HPPD was confirmed by the association between the concentration of those proteins and the length of AR CAG repeats.

The ALDOB is a glycolytic enzyme, predominantly expressed in liver and kidney, that catalyzes the reversible cleavage of fructose-1,6-bisphosphate into glyceraldehyde 3-phosphate and

**Table 3.** Ratio between mean concentrations of 4-hydroxyphenylpyruvate dioxygenase (4HPPD) and fructose-bisphosphate aldolase (ALDOB) in men with CAG repeat length <21 or > 22 as compared to the reference group.

| Proteins | **Overall** p value | <21 vs reference | | >22 vs reference | |
| --- | --- | --- | --- | --- | --- |
| | | Ratio (95% CI) | p value* | Ratio (95% CI) | p value* |
| 4HPPD | 0.012 | 1.34 (1.02–1.76) | 0.032 | 1.62 (1.23–2.13) | 0.001 |
| ALDOB | 0.008 | 1.35 (1.03–1.78) | 0.029 | 1.72 (1.29–2.32) | <0.001 |

*p value of the post hoc constrain between groups.
Ratio: ratio between mean concentration in <21 or >22 groups divided by the reference group (21 and 22).

dihydroxyacetone phosphate. The B isoform of aldolase, for example ALDOB, in the liver is under dietary control (*Munnich et al., 1985*). Ingestion of fructose induces ALDOB mRNA expression in the liver, which is otherwise low in fasting conditions. In humans, the absence of functional ALDOB enzyme due to mutations in the ALDOB gene cause hereditary fructose intolerance, characterized by metabolic disturbances that include hypoglycemia, lactic acidosis, and hypophosphatemia (*Hannou et al., 2018*). An upregulation of ALDOB in human pancreatic β cells occurs upon the development of hyperglycemia and may contribute to the impairment of insulin secretion in humans (*Gerst et al., 2018*). In a study on goats, the ALDOB gene was found to be downregulated at the time of postnatal initiation of spermatogenesis (*Bo et al., 2020*). This finding is in accordance with our data showing that rising testosterone is inhibiting ALDOB.

Similar to ALDOB, we found that 4HPPD was negatively associated with T levels. This enzyme is involved in the catabolic pathway of tyrosine and catalyzes the conversion of 4-hydroxyphenylpyruvate to homogentisic acid in the tyrosine catabolism pathway (*Hager et al., 1957*). The expression of the gene is regulated by hepatocyte-specific and liver-enriched transcription factors, as well as by hormones (*Kim et al., 2014*). Tyrosine has previously been reported to be upregulated in hypogonadal men and both tyrosine and phenylalanine levels were suggested as predictors of the risk of developing diabetes many years before manifest disease (*Wang et al., 2011*; *Fanelli et al., 2018*; *Guasch-Ferré et al., 2016*). In male tyrosine hydroxylase knockout mice normal body weight, puberty onset, and basal gonadotropin levels in adulthood were evident, although T was significantly elevated in adult mice (*Stephens et al., 2017*).

The last marker, IGFBP6 is expressed in most tissues (*Kim et al., 2014*) and is one of the binding proteins for insulin-like growth factor (IGF). The principal function of IGFBP6 is inhibiting IGF-II actions, whereby IGF-II-induced cell proliferation, differentiation, migration, and survival is reduced. Serum levels of IGFBP6 increase gradually with age and are higher in men than in women, but there are conflicting studies of the direct effects of sex steroids on IGFBP6 expression in different tissues (*Bach et al., 2013*). A positive association between the levels of T and IGFBP6 have previously been found (*Rooman et al., 2005*; *Huang et al., 2019*). The latter study has a somewhat similar set up as the present study, based on chemical castration with a GnRH agonist, and has also focused on identifying novel markers of BAA, but with candidate markers previously identified as being associated with changes in fat-free mass. The study showed that early increases in IGFBP6 levels in men receiving testosterone were associated with increases in fat-free mass and muscle strength.

Altogether, our findings may not only be clinically valuable in developing new methods of assessing BAA but also add to our understanding of the biological role of T in human metabolism, regulation of testicular function (ALDOB), as well as muscle strength and body composition (IGFBP6). However, our findings cannot be used as a direct proof of hypogonadism being cause of cardiometabolic disease but the combined parameter MMA may in this context be an important tool in the detection of long-term morbidity, such as bone mineral density and cardiometabolic risk, even before clinical diagnosis.

Our study has some strengths and limitations. Using a chemical castration model in young healthy men, we were able to identify proteins influenced by androgens and select those that were most strongly associated with T levels. By utilizing proteomics, we had an explorative approach to identify new markers of BAA without being restricted by previously published findings.

Another strength is the depth of the analysis due to the depletion abundant proteins from plasma. We were able to identify more than 450 proteins, which were identified in the same concentration range as 87% of FDA-approved biomarkers (*Anderson, 2010*). If depletion is not performed the detection level is dampened by the components from the digested abundant proteins as the proteins removed are of highest abundance in plasma and plasma proteome and exceed some lower abundance proteins by 10 orders of magnitude (*Tu et al., 2010*).

Although more clinical testing is needed, we have provided preliminary results showing that these protein markers may also be clinically useful. We have previously shown that median length CAG number is associated with most active AR (*Nenonen et al., 2010*). Thus, the fact that we find the lowest ALDOB and 4HPPD levels in those subjects having AR CAG repeat length close to median, confirms thatthe candidate markers identified in the present study are androgen dependent.

A limitation of our study is that the lack of reliable criteria for clinical hypogonadism, which made it impossible to test the power of the new markers in men in whom androgen replacement is needed. Furthermore, the clinical part of the study was limited, because we do not have sufficient information

about potential factors influencing the inter- and intraindividual variation in the levels of these proteins and, thereby, their suitability as clinical markers. Furthermore, the number of subjects included in the BL-T group was not sufficient to clarify whether, in this testosterone concentration interval, the new markers can be useful in discriminating between truly hypogonadal and men being eugonadal.

Furthermore, we are not reporting absolute values of quantifications for the potential markers, but the relative quantifications for comparing the protein expression between groups. This kind of comparative proteomics is favorable in research studies, in which preliminary results of protein changes between groups are obtained. Also, the sample processing is complicated putting high demands on the laboratory. In proteomics, there can be a high variation between laboratories in reporting absolute concentration proteins in plasma, especially when small sample sizes are reported (*Nanjappa et al., 2014*). In order to obtain trustworthy absolute concentration ranges or determine the activity level of the enzymes, it is necessary to analyze the potential markers in large cohorts including both healthy subjects and patients.

In this study, we have applied an immunoassay for measuring T levels, although some consider liquid chromatography–tandem mass spectrometry (LC–MS/MS) as gold standard in assessment of sex hormone levels. However, worldwide the former is most commonly used for T measurements. Additionally, in the concentration range seen in males, there seems to be high correlation between concentration values obtained by immunoassay and by LC–MS/MS (*Huhtaniemi et al., 2012*). Also in identifying men in hypogonadal T range and prediction of cardiometabolic risk, assessment of T by LC–MS/MS was not shown to be superior to that performed by standard methodology (*Huhtaniemi et al., 2012*; *Haring et al., 2013*).

For this study, the *z*-score was employed because of the relatively young age of the subjects. *z*-Scores, a comparison of an individual's bone density with that of a healthy reference population (NHANES III) of the same age, sex, and ethnicity and expressed as standard deviations (SDs), were obtained from the DXA machine. In this study, we defined low BMD as *z*-score below −1.0. The rationale is based primarily on meta-analysis of 12 cohort studies demonstrating significantly increased risk of osteoporotic fractures for men at *z*-scores ≤ −1 SD (*Johnell et al., 2005*) and in addition because it has also been shown that the majority of fragility fractures occur in patients with BMD in the osteopenic range, that is *T*-score between −1 and −2.5. (*Unnanuntana et al., 2010*). Based on this information, *z*-score below −1 can be assumed to imply an increased fracture risk.

In conclusion, we have identified three new potential biomarkers of BAA. Those proteins – alone or in combination – are promising as useful parameters in the clinical diagnosis of male hypogonadism and in the prediction of its long-term sequelae, as well as in studying the biology of androgen action. More extensive testing is vital to elucidate their BAA potential, not only in men but also in women and in prepubertal boys.

## Data availability

The mass spectrometry proteomics data have been deposited to the ProteomeXchange Consortium via the PRIDE (*Perez-Riverol et al., 2019*) partner repository with the dataset identifier PXD024448. Supplementary tables (datasets) https://doi.org/10.6084/m9.figshare.14875431 Source data of the Figures can be found on: https://doi.org/10/6084/m9.figshare.14875431 Supplementary Figure S1 (Figure2-figure supplement 1): https://doi.org/10.6084/m9.figshare.14876562. R code: https://github.com/indirapla/TP1_proteins_marker_of_androgen_activity, (copy archived at swh:1:rev:2613c2709a-14dec63c727d01edefb5a5f1f1fdc5; *Parada, 2022*).

## Additional information

### Competing interests
Aleksander Giwercman: received consulting fees from Besins Healthcare. The author has no other competing interests to declare. The other authors declare that no competing interests exist.

## Funding

| Funder | Grant reference number | Author |
|---|---|---|
| ReproUnion | 20201846 | Aleksander Giwercman |
| Swedish Governmental Fund for Clinical Research | | Aleksander Giwercman |

The funders had no role in study design, data collection, and interpretation, or the decision to submit the work for publication.

## Author contributions

Aleksander Giwercman, Conceptualization, Data curation, Methodology, Project administration, Writing – original draft, Writing – review and editing; K Barbara Sahlin, Data curation, Formal analysis, Investigation, Visualization, Writing – original draft, Writing – review and editing; Indira Pla Parada, Data curation, Formal analysis, Investigation, Methodology, Validation, Visualization, Writing – original draft, Writing – review and editing; Krzysztof Pawlowski, Data curation, Formal analysis, Software, Validation, Writing – review and editing; Carl Fehninger, Data curation, Formal analysis, Validation, Writing – review and editing; Yvonne Lundberg Giwercman, Conceptualization, Investigation, Validation, Writing – review and editing; Irene Leijonhufvud, Data curation, Investigation, Writing – review and editing; Roger Appelqvist, Data curation, Formal analysis, Funding acquisition, Investigation, Writing – review and editing; György Marko-Varga, Data curation, Funding acquisition, Methodology, Project administration, Supervision, Writing – review and editing; Aniel Sanchez, Data curation, Formal analysis, Investigation, Methodology, Validation, Visualization, Writing – original draft; Johan Malm, Conceptualization, Data curation, Formal analysis, Funding acquisition, Investigation, Methodology, Project administration, Supervision, Validation, Visualization, Writing – original draft, Writing – review and editing

## Author ORCIDs

Aleksander Giwercman (iD) http://orcid.org/0000-0001-5816-0785
Krzysztof Pawlowski (iD) http://orcid.org/0000-0002-5367-0935
Carl Fehninger (iD) http://orcid.org/0000-0003-0922-7749

## Ethics

All subjects were enrolled with informed written consent. The two studies from which they were recruited were approved by the Swedish Ethical Review Authority (approval number: DNR 2014/311, date of approval 8 May 2014; DNR 2011/1, date of approval 11 January 2011).

## Decision letter and Author response

Decision letter https://doi.org/10.7554/eLife.74638.sa1
Author response https://doi.org/10.7554/eLife.74638.sa2

---

# Additional files

## Supplementary files

• Transparent reporting form

## AleksanderGData availability

The mass spectrometry proteomics data have been deposited to the ProteomeXchange Consortium via the PRIDE (60) partner repository with the dataset identifier PXD024448. Supplementary tables (datasets): https://doi.org/10.6084/m9.figshare.14875431 Source data of the Figures can be found on: https://doi.org/10.6084/m9.figshare.14875431 Supplementary Figure S1 (Figure 2-figure supplement 1): https://doi.org/10.6084/m9.figshare.14876562.

The following datasets were generated:

| Author(s) | Year | Dataset title | Dataset URL | Database and Identifier |
|---|---|---|---|---|
| Giwercman A, Sahlin KB, Pla I, Pawlowski K, Giwercman YL, Leijonhufvud I, Appelqvist R, Sanchez A, Malm J | 2021 | Supplementary tables, Novel protein markers of androgen activity in humans with potential clinical value | https://doi.org/10.6084/m9.figshare.14875431.v2 | figshare, 10.6084/m9.figshare.14875431.v2 |
| Giwercman A, Sahlin KB, Pla I, Pawlowski K, Giwercman YL, Leijonhufvud I, Appelqvist R, Sanchez A, Malm J | 2021 | Supplementary Figure S1, Novel protein markers of androgen activity in humans with potential clinical value | https://doi.org/10.6084/m9.figshare.14876562.v1 | figshare, 10.6084/m9.figshare.14876562.v1 |
| Giwercman A, Sahlin KB, Pla I, Pawlowski K, Giwercman LY, Leijonhufvud I | 2022 | Novel potentially clinically valuable protein markers of androgen activity in humans | https://www.ebi.ac.uk/pride/archive/projects/PXD024448 | PRIDE, PXD024448 |

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

# APPENDIX 1

# Supplementary description of methods

## Clinical chemistry parameters, bone mineral densityand CAG-repeat length

### Laboratory tests and dual X-ray absorptiometry

The analysis of the clinical chemistry parameters as well as bone mineral density were conducted as previously described *Bobjer et al., 2016* in the study on infertile men. Markers measured in fasting morning plasma samples included: total T, luteinizing hormone (LH), follicle stimulating hormone (FSH), Sex hormone-binding globulin(SHGB), estradiol (E2), glucose, insulin, apolipoprotein A1, and apolipoprotein B. The parameters were analyzed at the Department of clinical chemistry, Skåne University Hospital, Malmö, Sweden. Total testosterone, LH, FSH and SHGBwere measured by electrochemiluminescence immunoassay (ECLIA) and an immunofluorometric method (DELFIA Estradiol, Wallac OY, Finland) was utilized to measure E2. Glucose was measured by an automated hexokinase method and insulin was measured by immunometric sandwich assay. Bone mineral density (BMD, g/cm²) at total hip (TH) and lumbar spine L1–L4 (LS) was measured using dual X-ray absorptiometry.

### CAG-repeat length

Androgen receptor CAG-repeat length was determined in all the infertile men. For the analysis, DNA was extracted from peripheral leukocytes. The androgen receptor CAG stretch was amplified by polymerase chain reaction (PCR), purified and directly sequenced in a Beckman Coulter CEQ 2000 XL (Beckman Coulter, Bromma, Sweden) as previously described *Lundin et al., 2003*.

## Proteomics experiments

All chemical reagents for the proteomic analysis not specified in the text were purchased from Sigma Aldrich (St. Louis, MO). Modified porcine trypsin was from Promega (Madison, WI), water from Milli-Q Ultrapure Water system (Millipore, Billerica, MA). Water and organic solvents for LC-MS were of LC-MS quality (Merck, Darmstadt, Germany).

The plasma and serum samples from respective cohort (healthy men and infertile patients) were randomized regarding the order before starting sample preparation. Quantitation of total protein content in the samples was performed using the bicinchoninic acid (BCA) assay (*Smith et al., 1985*). Following the instructions provided by the manufacturer, the top seven most-abundant proteins (albumin, IgG, IgA, transferrin, haptoglobin, antitrypsin and fibrinogen) were depleted from approximately 10 μL of each sample using a MARS7 column (Agilent, Santa Clara, CA). The buffer was exchanged to 1.6% SDC, 50 mM $NH_4HCO_3$ with Amicon ultra-centrifugal filters (0.5 mL, 10 kDa cut-off, Millipore, Tullagreen, Ireland). The disulphide bonds were reduced by adding DTT to a final concentration of 10 mM and incubated 1 h at 37 °C. The free thiol groups were alkylated by adding iodoacetamide to a final concentration of 25 mM, and the reaction proceeded for an additional 30 min at room temperature in darkness (reduction and alkylation occurred in the Amicon filter). The buffer was exchanged to 50 mM $NH_4HCO_3$ and the samples were resuspended in 100 μL 50 mM $NH_4HCO_3$ (30 μg protein after BCA quantitation) and digested with trypsin at an enzyme-to-substrate mass ratio of 1:30 (w/w) for 16 h at 37 °C. The remaining SDC was precipitated by adding 20% formic acid prior to filtering the samples through a polypropylene filter plate with a hydrophilic PVDF membrane (mean pore size 0.45 μm, Porvair Filtration Group, Fareham, UK).

A Q-Exactive Plus mass spectrometer connected to an Easy-nLC 1000 liquid chromatography system (Thermo Scientific, San José, CA, USA) was used to analyse the samples for automated Data Dependent Acquisition (DDA) methods. Peptides (2 μL, 1 μg on the column) were loaded onto an Acclaim PepMap 100 precolumn (75 μm × 2 cm, Thermo Scientific, San José, CA, USA) and separated on an easy-Spray column (25 cm ×75 μm i.d., PepMap C18 2 μm, 100 Å) at a flow rate of 300 nL/min and a column temperature of 35 °C. Solvent A (0.1% formic acid) and solvent B (0.1% formic acid in acetonitrile) were used to create a nonlinear gradient to elute the peptides (90 min).

The MS1 spectra of the peptides were acquired in the orbitrap mass analyser from *m/z* 400–1,600 and a resolution of 70,000 (at *m/z* 200). The target automated gain control (AGC) value and maximum injection time (IT) were 1e6 and 100ms, respectively. The ten most intense peaks with charge state ≥2 were fragmented by higher-energy collisional dissociation (HCD) with 26% normalised collision energy. Tandem mass spectra were acquired in the orbitrap mass analyser at

a resolution of 35,000 (at *m/z* 200), a target AGC value of 5e4 and a maximum IT of 100ms. The underfill ratio was 10% and dynamic exclusion was 45 s. The normalised collision energy was 26%.

Selected proteins from the heatlhy individuals were monitored also in patients by Multiple Reaction Monitoring (MRM) method. A TSQ Quantiva mass spectrometer equipped with an EASY-Spray NG ion source and connected to an EASY n-LC 1000 liquid chromatography system (Thermo Scientific, San José, CA, USA) was used for the MRM method. Peptides (5 µL, 1 µg on the column) were loaded onto an Acclaim PepMap 100 precolumn (100 µm × 2 cm, Thermo Scientific, San José, CA, USA) and separated on an easy-Spray column (15 cm ×75 µm i.d., PepMap C18 3 µm, 100 Å) at a flow rate of 300 nL/min and a column temperature of 35 °C. Solvent A (0.1% formic acid) and solvent B (0.1% formic acid in acetonitrile) were used to create a nonlinear gradient to elute the peptides (60 min). Internal references, isotopically labelled synthetic peptides (UVic-Genome BC Proteomics Centre, BC, Canada) were added in order to determine relative quantifications of proteins. Data was acquired in scheduled MRM mode with 5 min detection windows. SRM transitions were acquired in Q1 and Q3 operated at unit resolution (0.7 full witdth at half maximun, FWHM), the collision gas pressure in Q2 was 1.5 mTorr. The cycle time was 2 s, and calibrated radio frequency (RF) and S-lens values were used. At least three transition per precursor were monitored and in most cases the ratio is reported between the relative total intensities vs the references.

For peptide and protein identification the Proteome Discoverer v2.2 (PD) software (Thermo Scientific, San José, CA) was used. Peptides were identified using SEQUEST HT against UniProt (http://www.uniprot.org) human database (Human 9606, Reviewed, 20 165) integrated into PD. The search was performed with the following parameters: cysteine carbamidomethylation as a static modification, oxidation of methionine as a dynamic modification, 10 ppm precursor tolerance and 0.02 Da fragment tolerance. Up to one missed cleavage for tryptic peptides was allowed. According to the PD software, the filters applied were high- (FDR < 0.01) and medium confidence (FDR < 0.05) at peptide- and protein levels, respectively. The peptide and protein quantifications were based on the quantification of MS peptide signals (label-free quantification). Label-free quantification used the Minora Feature Detector node in the processing workflow, and the Precursor Ions Quantifier node and the Feature Mapper in the consensus workflow.

To analyse the MRM data, all MS files were imported into Skyline v3.5 (17) (MacCoss Lab Software, Seattle, WA). The peak integration was performed automatically by the software, using Savitzky-Golay smoothing and to confirm the correct peak detection it was in addition, manually inspected. The relative total intensities vs the references (ratios), were reported as expression values. Only the values that passedthe inspected intensity selection criteria according to PD or in Skyline were reported.

## Supplementary Statistical analyses

### Identification of candidate marker proteins from chemically castrated men

Proteins identified by mass spectrometry techniques in samples from healthy subjects were pre-processed in Perseus v1.6.7.0 (18) software. Intensities were Log2 transformed to follow a normal distribution and standardized by subtracting the median of its respective sample from each value. The statistical analyses were performed using R software (, *R Development Core Team, 2016*) and SPSS Statistics 21.0 (IBM, Somers, IL, USA). Differentially expressed proteins were determined by doing one-way repeated measures ANOVA (R function: ezANOVA{ez}) to reveal overall differences between conditions (normal (A), low (B) and restored (C) T) and differences between individual conditions were detected by performing a post-hoc test based on pairwise t-test (two-tails and paired) (R function: pairwise.t.test{stats}). The 'pairwise.t.test' function utilized the proteins with significant overall changes (ANOVA *P*-value < 0.05) to perform pairwise comparisons between conditions while corrected for multiple pairwise testing. Proteins with adjusted *p*-values ('*fdr*' method) < 0.05 following the pairwise t-test were considered significant.

Because gonadotropins (FSH, LH) did not change their levels between conditions B and C, while T did recover its value (*Sahlin et al., 2020*), proteins that changed with statistical significance between A-B and recovered their values significantly between B and C were considered mainly influenced by T changes and, hence, were selected for further analyses.

Based on the outcome of the first selection step of the analysis, we performed receiver operating characteristic (ROC) analysis to select proteins capable to discriminate between normal and low T. Two categories of samples were created: 'low T' that included samples evaluated at time point

B and 'normal T' that included samples evaluated at time point A and C. Although there is no consensus about the threshold for sufficient diagnostic accuracy, the area under the curve (AUC) can be interpreted as the following rough classifying system: > 0.90 = excellent; 0.80–0.90 = very good; > 0.70–0.80 = good; 0.60–0.70 = sufficient and 0.50–0.60 = poor (*Marshall et al., 2010*, *Simundić, 2009*). Consequently, based on the ROC-AUC values, we created two cut-off values to select the final protein biomarkers: (a) AUC >0.80 (i.e very good) and (b) AUC 0.75–0.80 (i.e. good) and highly enriched in liver tissues according to the Human Proteome Map (*Kim et al., 2014*) and (*Kampf et al., 2014*), which, unlike other organs, appears to be relatively protected from age-related changes (*Kholodenko and Yarygin, 2017*; *Schmucker and Sanchez, 2011*). The Human Proteome Map tool (*Kim et al., 2014*) was used to determine the tissue-specific localization of the proteins.

Because the combination of different markers may improve the discriminative power to diagnose hypogonadism and predict its long term sequelae, proteins selected from the ROC-AUC analysis were included as predictors in a stepwise regression (method: backward) to select the best combination of markers that predict the odds of being low T. Bootstrap resampling with replacement method was applied to assess consistency of predictors selected with the stepwise regression. A new variable called Multi Marker Algorithm (MMA) was derived from the predicted log-odds (of being low T) obtained from a binomial logistic regression analysis, in which the dichotomized variable 'testosterone level' (0: normal T (time points A, C) and 1: low T (time point B)) was used as the dependent variable and the intensities of the selected proteins from the stepwise regressionwere the predictors' variables. Finally, MMA variable was evaluated together with marker candidates proteins. Subsequently, an ROC analysis was performed based on the predicted values. All analyses described in this part were performed in R software and plotted using ggroc{pROC} function.

## Background characteristics of the infertile cohort of patients

To describe background characteristics of the infertile cohort of patients (*Table 1*), the mean and standard deviation (SD) was calculated for variables that followed a normal distribution or the median and minimum-maximum (min-max) for variables that were not normally distributed (non-Gaussian). The normal distribution was checked by applying a Kolmogorov-Smirnov test.

## Testing of the candidate biomarkers in infertile men

To further perform statistical analyses, the quantified intensities of the candidate biomarkers in infertile men were Log2 transformed to achieve normal distributions. Markers selected from the stepwise regression in the healthy human model were also considered in this cohort to create the MMA variable. In this case, MMA was created to predict the odds of suffering low T or other medical conditions associated with low T levels. Unless other software is specified, the statistical analyzes described in the next steps were performed using R software (*RStudio Team, 2016*, *R Development Core Team, 2016*).

## Distinguishing infertile men with different testosterone levels

In order to know if the change in the proteins occur with the change in T as observed in the healthy human model, we created three groups of patients based on total T concentration values commonly described in clinical guidelines (*Diaz-Arjonilla et al., 2009*) to define T deficiency: Group one contains patients commonly defined with low T (LT): ≤ 8 nmol/L (n = 22), Group two contains patients defined with borderline low T (BL_T): between 8 and 12 nmol/L (n = 17), and Group three contains patients with normal T (NT): > 12 nmol/L (n = 36). Differences among the three groups of patients were detected by doing a one-way ANOVA per protein (outcome variable: protein intensity) followed by a pairwise analysis that was corrected to control the FDR provoked by multiple pairwise comparisons (Two-stage linear step-up procedure of *Benjamini et al., 2006*, performed in GraphPad Prism software, version 9.00 for Windows, CA, USA). Adjusted $P$-values < 0.05 were considered significant. Calculated free testosterone (cFT) was determined according to the method described by *Vermeulen et al., 1999*. The cut-off level of 220 pmol/L was used to categorize the subjects as having lowcFT (L_cFT; n = 21) or normal cFT(N_cFT; n = 54) (*Antonio et al., 2016*). Changes between the two groups were evaluated bytwo-tailed Student's T-test ($P$-values < 0.05 were considered significant).

An ROC analysis was performed to discriminate the patients with LT from those with BL_T or NT. The same analysis was repeated after excluding the BL_T patients as this group includes males in the

gray zone for hypogonadism diagnosis (*Chan et al., 2014*; *Zitzmann et al., 2006*). ROC analysis was also used to discriminate between L_cFT and N_cFT.

## Distinguishing between men with abnormal cardio-metabolic parameters or reduced bone mineral density

In order to evaluate the diagnostic value of our new protein markers in relation to conditions associated with low T levels, we performed a ROC analysis to discriminate patients with MetS, IR, CVRLP, DM or LBD within the patient cohort using the expression level of the candidate biomarkers and T. Based on the ROC curves, the DeLong's test (paired) was applied in R (roc.test {pROC} function) to compare the AUCs of the candidate biomarkers vs the AUC of T levels to discriminate the above mentioned pathological conditions.

## Association with Androgen receptor CAG-repeat length

To investigate the androgenic dependence of the levels of the newly defined protein markers, we looked for associations between CAG repeat length and concentration of those markers. The infertile cohort was separated into three groups based on CAG repeat length of the androgen receptor (AR); a group of patients with CAG repeat length 21 and 22 (n = 18) was defined as reference group (*Casella et al., 2001*; *Ferlin et al., 2004*). Patients with CAG repeat length less than the reference group ( < 21) were included in a group (n = 26) and patients with CAG repeat length longer than the reference group ( > 22) made up the third group (n = 30). Using the softwere GraphPad Prism version 9.0.0 for Windows (CA, USA), a one-way ANOVA followed by a pairwise test (two-stage linear step-up procedure of *Benjamini et al., 2006*) which controls FDR for multiple pairwise comparison was conducted and adjusted $P$-values < 0.05 were considered significant.

