## [Editor Report]

This work by Giwercman, et al., interrogates the identity of potential protein biomarkers of androgen activity in humans by carrying out a proteomic analysis in blood from 30 healthy males treated at baseline, after medical castration and at a third time point after testosterone replacement. Proteins most significantly associated with testosterone changes were tested further in a separate cohort. The major findings include the observation that 4 specific proteins are potential protein biomarkers that follow testosterone levels and presumably androgen receptor activity, thus providing new insights into androgen physiology and pathophysiology.

---

## [Decision Letter]

**Decision letter after peer review:**

Thank you for submitting your article "Novel protein markers of androgen activity in humans: proteomic study of plasma from young chemically castrated men" for consideration by *eLife*. Your article has been reviewed by 2 peer reviewers, one of whom is a member of our Board of Reviewing Editors, and the evaluation has been overseen by a Reviewing Editor and Nancy Carrasco as the Senior Editor. The reviewers have opted to remain anonymous.

Essential revisions:

1) Testosterone measurements were performed using an immunoassay, and not using the gold standard technique for these measurements which is by mass spectrometry. If one wants to define other/superior markers for androgen activity than testosterone itself, this should be done by comparing this using the gold standard technique of the measurement.

2) As the authors describe in the introduction, SHBG levels are of major importance while interpreting testosterone results in hypogonadal men. It is believed that the free or bioavailable fraction represents the true amount of active hormone. In searching for other markers of androgen activity, these should at least be compared to the free and/or bioavailable levels of the hormone (either measured or calculated using a validated formula).

3) The authors divide the group of infertile men into 3 categories: LT (T {less than or equal to} 8 nmol/L), BL_T (T 8 -12 nmol/L) and NT ({greater than or equal to} 12 nmol/L). This is indeed interesting, because patients in the LT group are highly likely to be androgen deficient, while patients in the NT group are highly likely to have sufficient androgen activity. The issue, as correctly addressed by the authors, situates itself in the BL_T group, where the interpretation of the T levels is difficult, and symptoms described by the patient, which are often rather general or possibly multifactorial (fatigue, erectile dysfunction, decrease libido) are uncertain to be linked to the borderline low T values. It would be therefore of interest, if other biomarkers would be able to differentiate between 'low' and 'normal' especially in this borderline low range, where a lot of uncertainty exist. However, when looking at figure 3a, the selected biomarkers are indeed good in differentiating between LT and NT, and LT and BL_T, but they do not seem to be able to discriminate between BL_T and NT levels (not statistically significantly different). So, when evaluating the patients in the BL_T range by determining these new biomarkers (4HPPD, IGFBP6, ALDOB and MMA), one should conclude that they are not androgen deficient. These markers hence do not provide an added value on top of solely measuring T levels, and this does not resolve the issue as described above, as they are not able to discriminate between BL_T and NT.

4) The use of androgen deficiency linked pathologies to validate new biomarkers of androgen activity has important limitations. The pathologies suggested by the authors have a clear association with androgen deficiency, but these associations have certainly not yet been proven to be causal and probably aren't. If the association between these biomarkers and the pathologies are stronger than the association with testosterone levels, this thus not mean that these markers are better biomarkers of androgen activity than testosterone itself, but may just be better markers of the suggested pathology itself, apart from any possible link with androgen action. It would be therefore much more useful to assess symptoms of hypogonadism (such as low libido and erectile dysfunction), and link these new biomarkers with these symptoms which are known to be caused by androgen deficiency. This is a major limitation for the interpretation of the results.

5) Table 3, which is the ratio between mean concentrations of 4HPPD and ALDOB in men with various CAG repeat lengths, is not very well described in the text. It seems cursory and really an afterthought of the story. Why use these specific numbers and thresholds (CAG repeat length of 21, 22), as opposed to other thresholds? Is this justified by something in the literature? This should be clarified and justified.

6) Compared with the other data on associations between T and protein biomarkers, the CAG repeat data seem to be relatively weak and I worry it might distract from the other strengths in this paper. Other data on CAG and prostate cancer, for example, use thresholds of 18 and 26 CAG repeats (e.g., Giovannucci, et al., PNAS 1997).

7) Similarly, the hypothesis for the data in Figure 5 and Table 3 is not clear. This should really be clarified.

*Reviewer #2 (Recommendations for the authors):*

This work reveals associations of some protein levels in circulation with androgen status and a first validation in a hypoandrogenism cohort.

*Reviewer #3 (Recommendations for the authors):*

Major comments:

1) Table 3, which is the ratio between mean concentrations of 4HPPD and ALDOB in men with various CAG repeat lengths, is not very well described in the text. It seems cursory and really an afterthought of the story. Why use these specific numbers and thresholds (CAG repeat length of 21, 22), as opposed to other thresholds? Is this justified by something in the literature? This should be clarified and justified.

2) Compared with the other data on associations between T and protein biomarkers, the CAG repeat data seem to be relatively weak and I worry it might distract from the other strengths in this paper. Other data on CAG and prostate cancer, for example, use thresholds of 18 and 26 CAG repeats (e.g., Giovannucci, et al., PNAS 1997).

3) Similarly, the hypothesis for the data in Figure 5 and Table 3 is not clear. This should really be clarified.

---

## [Author Response]

Essential revisions:1) Testosterone measurements were performed using an immunoassay, and not using the gold standard technique for these measurements which is by mass spectrometry. If one wants to define other/superior markers for androgen activity than testosterone itself, this should be done by comparing this using the gold standard technique of the measurement.

We are aware that use of LC-MS/MS has been suggested as a new gold standard in measuring testosterone levels. However, the access to this method is still very limited and the vast majority of clinical analyses are still performed using immunoassays. A number of studies, including (Huhtaniemi et al., 2012), have demonstrated that in the testosterone concentration range found in males, there is a high correlation between the concentration measured with immunoassays and by LC-MS/MS. Immunoassays are also reliable in identifying men with low testosterone levels. Thus, the superiority of LC-MS/MS is mostly limited to measurements done in prepubertal boys and women, who typically have very low levels of testosterone.

This issue was already addressed in the Discussion part of the first version of our manuscript, but has now been extended by following (Line 362-365): “Additionally, in the concentration range seen in males, there seems to be high correlation between the measurements of serum concentrations obtained by immunoassay and by LC-MS/MS (Huhtaniemi et al., 2012). Also in identifying men in hypogonadal testosterone range and in prediction of cardiometabolic risk, assessment of testosterone by LC-MS/MS was not superior to that performed by standard methodology (Huhtaniemi et al., 2012; Haring et al., 2013).

2) As the authors describe in the introduction, SHBG levels are of major importance while interpreting testosterone results in hypogonadal men. It is believed that the free or bioavailable fraction represents the true amount of active hormone. In searching for other markers of androgen activity, these should at least be compared to the free and/or bioavailable levels of the hormone (either measured or calculated using a validated formula).

The reason why we have focused on total testosterone only is because this type of measurement is most commonly used in clinical practice, whereas the use of measured or calculated free testosterone is still very limited and more of an academic matter. However, following the suggestion of the reviewers and editors, we have added data on calculated free testosterone using the formula by Vermeulen (Vermeulen et al., 1999). These results, now added to Table 2 and Figure 3, show that the protein markers reported by us, are almost as reliable in defining low free testosterone as they are for total testosterone.

3) The authors divide the group of infertile men into 3 categories: LT (T {less than or equal to} 8 nmol/L), BL_T (T 8 -12 nmol/L) and NT ({greater than or equal to} 12 nmol/L). This is indeed interesting, because patients in the LT group are highly likely to be androgen deficient, while patients in the NT group are highly likely to have sufficient androgen activity. The issue, as correctly addressed by the authors, situates itself in the BL_T group, where the interpretation of the T levels is difficult, and symptoms described by the patient, which are often rather general or possibly multifactorial (fatigue, erectile dysfunction, decrease libido) are uncertain to be linked to the borderline low T values. It would be therefore of interest, if other biomarkers would be able to differentiate between ‘low’ and ‘normal’ especially in this borderline low range, where a lot of uncertainty exist. However, when looking at figure 3a, the selected biomarkers are indeed good in differentiating between LT and NT, and LT and BL_T, but they do not seem to be able to discriminate between BL_T and NT levels (not statistically significantly different). So, when evaluating the patients in the BL_T range by determining these new biomarkers (4HPPD, IGFBP6, ALDOB and MMA), one should conclude that they are not androgen deficient. These markers hence do not provide an added value on top of solely measuring T levels, and this does not resolve the issue as described above, as they are not able to discriminate between BL_T and NT.

Thank you very much for this interesting comment. We agree that our markers cannot discriminate between the BL-T and NT groups. However, we do not think that present data exclude that the new markers may be superior to testosterone measurements in diagnosing men with testosterone deficiency. It is obvious that the BL_T group contains a mixture of men who are hypogonadal and those being eugonadal. Therefore, more data are needed to clarify if any of the presented protein markers can be used to discriminate, within the BL_T group, between those being truly hypogonadal and those being eugonadal. However, this question cannot be answered within the framework of current study. We have, therefore, added the following sentence to the Discussion (Line 346-349): “Furthermore, the number of subjects included in the BL-T group was not sufficient to clarify whether, in this testosterone concentration interval, the new markers can be useful in discrimination between truly hypogonadal men and men being eugonadal.”

4) The use of androgen deficiency linked pathologies to validate new biomarkers of androgen activity has important limitations. The pathologies suggested by the authors have a clear association with androgen deficiency, but these associations have certainly not yet been proven to be causal and probably aren’t. If the association between these biomarkers and the pathologies are stronger than the association with testosterone levels, this thus not mean that these markers are better biomarkers of androgen activity than testosterone itself, but may just be better markers of the suggested pathology itself, apart from any possible link with androgen action. It would be therefore much more useful to assess symptoms of hypogonadism (such as low libido and erectile dysfunction), and link these new biomarkers with these symptoms which are known to be caused by androgen deficiency. This is a major limitation for the interpretation of the results.

We completely agree that the pathogenetic mechanism and the direction of association between cardiometabolic disease and hypogonadism is not fully clarified and can be bi-directional. As for the link between hypogonadism and decreased bone density, the causality, which mostly is related to estrogen action, seems clearer.

It was not our intention to claim that low testosterone levels are cause of cardiometabolic disease, but we find that including data on association between the levels of the new protein markers and cardiometabolic parameters adds to understanding the potential clinical relevance of our findings. We have, added the following clarifying sentence to the Discussion part (Line 322-325):

“However, our findings cannot be used as a direct proof of hypogonadism as the cause of cardio-metabolic disease, but the combined parameter MMA may in this context be an important tool in the detection of long-term morbidity, such as bone mineral density and cardio-metabolic risk, even before clinical diagnosis”.

As considers decrease in libido and erectile dysfunction, it is true that some studies, including European Male Aging Study, have found these parameters to be the best clinical markers of hypogonadism. Unfortunately, we do not have sufficient data to include these parameters in our analysis. Furthermore, even those conditions are multi-factorial why an association with the protein markers cannot be considered as a proof of an androgen-dependent effect.

5) Table 3, which is the ratio between mean concentrations of 4HPPD and ALDOB in men with various CAG repeat lengths, is not very well described in the text. It seems cursory and really an afterthought of the story. Why use these specific numbers and thresholds (CAG repeat length of 21, 22), as opposed to other thresholds? Is this justified by something in the literature? This should be clarified and justified.

The following text describes the findings reported in Table 3 and Figure 5 (Line 262-266):

“Statistically significant inter-CAG-group overall differences were observed for 4HPPD (p = 0.012) and ALDOB (p = 0.008) (Figure 5). Additionally, the protein expressions were significantly higher in the groups with <21 and >22 CAG repeat length as compared with the reference (Table 3 and Figure 5). However, we did not observe any statistically significant association between CAG number and expression of IGFBP6.”

We recognise that the background for categorization of CAG lengths was not completely clarified in the first version of the manuscript. The selection of CAG lengths of 21 and 22 as reference was not random, but based on our previous in vitro and in vivo data showing that the CAG number of 22, corresponding to the mean length in our cohort, is associated with highest receptor activity. We have now added the following sentence in the Statistical analysis (Line 193-196):

“This categorization was undertaken in order to have three groups of sufficient size and the category including the mean CAG length value of 22 was chosen as reference since this CAG number was previously seen, in vitro and in vivo to be associated with highest receptor activity (27,28).”

6) Compared with the other data on associations between T and protein biomarkers, the CAG repeat data seem to be relatively weak and I worry it might distract from the other strengths in this paper. Other data on CAG and prostate cancer, for example, use thresholds of 18 and 26 CAG repeats (e.g., Giovannucci, et al., PNAS 1997).7) Similarly, the hypothesis for the data in Figure 5 and Table 3 is not clear. This should really be clarified.

In order to declare the rational for including data on androgen receptor CAG repeat lengths, we have added the following sentence to the statistical analysis section (Line 188-190):

“In order to strengthen the evidence of androgenic dependence of the candidate biomarkers, we looked for associations between their expression and the androgen receptor (AR) CAG repeat length, which was previously reported to have an impact on the activity of the receptor”.

We think that this part represents an important aspect of this manuscript. In the human model of GnRH-antagonist treated men, candidate markers were identified under conditions of significant changes in testosterone levels but also in the levels of other hormones, e.g. gonadotrophins. By showing that the levels of these protein markers are also dependent on more discrete and genetically determined variation in androgenic activity, as those associated with CAG number, we demonstrate their potential clinical value. We would appreciate to have these data kept in the manuscript but are willing to omit them in case the Reviewers and Editors consider it as crucial for accepting our paper.

References

Haring R, Baumeister SE, Nauck M, Volzke H, Keevil BG, Brabant G, Wallaschofski H. Testosterone and cardiometabolic risk in the general population – the impact of measurement method on risk associations: A comparative study between immunoassay and mass spectrometry. *Eur J Endocrinol* 2013;169:463–470.

Huhtaniemi IT, Tajar A, Lee DM, O’Neill TW, Finn JD, Bartfai G, Boonen S, Casanueva FF, Giwercman A, Han TS, *et al.* Comparison of serum testosterone and estradiol measurements in 3174 European men using platform immunoassay and mass spectrometry; relevance for the diagnostics in aging men. *Eur J Endocrinol* 2012;166:983–991.

Johnell O, Kanis JA, Oden A, Johansson H, Laet C De, Delmas P, Eisman JA, Fujiwara S, Kroger H, Mellstrom D, *et al.* Predictive Value of BMD for Hip and Other Fractures. *J Bone Miner Res* 2005;20:1185–1194.

Nenonen H, Björk C, Skjaerpe PA, Giwercman A, Rylander L, Svartberg J, Giwercman YL. CAG repeat number is not inversely associated with androgen receptor activity in vitro. *Mol Hum Reprod* 2010;16:153–157.

Nenonen HA, Giwercman A, Hallengren E, Giwercman YL. Non-linear association between androgen receptor CAG repeat length and risk of male subfertility – a meta-analysis. *Int J Androl* 2011;34:327–332.

Unnanuntana A, Gladnick BP, Donnelly E, Lane JM. The assessment of fracture risk. *J Bone Joint Surg Am* 2010;92:743–753.

Vermeulen A, Verdonck L, Kaufman JM. A critical evaluation of simple methods for the estimation of free testosterone in serum. *J Clin Endocrinol Metab* 1999;84:3666–3672.